# Identifying cell states in single-cell RNA-seq data at statistically maximal resolution

**Pascal Grobecker** [ID]**, Thomas Sakoparnig** [ID]**, Erik van Nimwegen** [ID]*

Biozentrum, University of Basel and Swiss Institute of Bioinformatics, Basel, Switzerland

* erik.vanimwegen@unibas.ch

**Data Availability Statement:** All data analyzed in this study are publicly available and links are provided to the raw data of each dataset studied in S1 Table. All data underlying the results shown are freely available from https://doi.org/10.5281/

## Abstract

Single-cell RNA sequencing (scRNA-seq) has become a popular experimental method to study variation of gene expression within a population of cells. However, obtaining an accurate picture of the diversity of distinct gene expression states that are present in a given dataset is highly challenging because of the sparsity of the scRNA-seq data and its inhomogeneous measurement noise properties. Although a vast number of different methods is applied in the literature for clustering cells into subsets with 'similar' expression profiles, these methods generally lack rigorously specified objectives, involve multiple complex layers of normalization, filtering, feature selection, dimensionality-reduction, employ *ad hoc* measures of distance or similarity between cells, often ignore the known measurement noise properties of scRNA-seq measurements, and include a large number of tunable parameters. Consequently, it is virtually impossible to assign concrete biophysical meaning to the clusterings that result from these methods. Here we address the following problem: Given raw unique molecule identifier (UMI) counts of an scRNA-seq dataset, partition the cells into subsets such that the gene expression states of the cells in each subset are statistically indistinguishable, and each subset corresponds to a distinct gene expression state. That is, we aim to partition cells so as to maximally reduce the complexity of the dataset without removing any of its meaningful structure. We show that, given the known measurement noise structure of scRNA-seq data, this problem is mathematically well-defined and derive its unique solution from first principles. We have implemented this solution in a tool called CELLSTATES which operates directly on the raw data and automatically determines the optimal partition and cluster number, with zero tunable parameters. We show that, on synthetic datasets, CELLSTATES almost perfectly recovers optimal partitions. On real data, CELLSTATES robustly identifies subtle substructure within groups of cells that are traditionally annotated as a common cell type. Moreover, we show that the diversity of gene expression states that CELLSTATES identifies systematically depends on the tissue of origin and not on technical features of the experiments such as the total number of cells and total UMI count per cell. In addition to the CELLSTATES tool we also provide a small toolbox of software to place the identified cellstates into a hierarchical tree of higher-order clusters, to identify the most important differentially expressed genes at each branch of this hierarchy, and to visualize these results.

zenodo.11577547 and the code used in the manuscript is freely available from https://github.com/nimwegenLab/cellstates.

**Funding:** This work was supported through grant number 310030_184937 of the Swiss National Science Foundation (https://www.snf.ch) to EvN. The funders had no role in study design, data collection and analysis, decision to publish, or preparation of the manuscript.

**Competing interests:** The authors have declared that no competing interests exist.

## Author summary

Single-cell RNA sequencing (scRNA-seq) has the promise to offer fundamental new insights into how gene expression is regulated but analyzing such data is very challenging because of its sparsity and heterogeneity in its measurement noise. For example, one common component of scRNA-seq analysis procedures is the clustering of groups of cells with similar gene expression, but current methods use complex *ad hoc* schemes that involve multiple layers of complex transformations of the data making it hard to interpret their results. Here we present a method that clusters cells so as maximally reduce the complexity of the data without removing any of its meaningful structure, by grouping only cells whose expression profiles are statistically indistinguishable. Importantly, we show that, given the known measurement noise structure of scRNA-seq data, this problem has a unique solution which we derive from first principles. We implemented this method in a tool called Cellstates which operates directly on the raw data with zero tunable parameters. We validate the power of Cellstates by showing it performs almost perfectly on synthetic data and outcompetes existing methods on real data consisting of known mixtures of cells.

## Introduction

All cells in multicellular organisms contain the same genome with typically around 20, 000 genes but are able to take on a wide variety of phenotypes and perform specialized functions by selective expression of these genes. Therefore, it is one of the fundamental problems of cell biology to characterize the gene expression states cells take on in a multicellular organism. Addressing this requires investigating gene expression states in single cells, which has become possible through progress in the development of single-cell technologies, and single-cell RNA sequencing (scRNA-seq) in particular, over the last years. Numerous cell atlas projects [1–5] using this approach are already published or in progress. It is often assumed that cells can be divided into discrete cell types which have characteristic molecular profiles and perform specific functions, but despite of all the available experimental data, it is still debated how such discrete types should be defined [6–9]. Indeed, it is also often proposed that gene expression states are not discrete but rather occupy a continuous subspace of gene expression space, which is typically assumed to be of much lower dimensionality than the full gene expression space [10, 11].

It is thus currently not clear to what extent the assumption that cells can be grouped into discrete states is appropriate. Arguably, during cellular differentiation cells must be traversing an approximately continuous space of gene expression states, but for fully differentiated tissues it may not be unreasonable to approximate cells as deriving from a set of discrete states. However, even if we take for granted the assumption that cells take on discrete states or 'types', there is currently also no agreement regarding how such types should be defined or identified. That is, although intuitively cells of the same type should have 'similar' expression profiles, there is currently no agreed upon metric of closeness of gene expression states and no agreement on how close cells need to be in order for them to be considered the same type. Furthermore, even if a distance metric is chosen, for example Euclidean distance in log mRNA fractions, the sparseness and inhomogeneous noise properties of scRNA-seq data make it very challenging to accurately estimate the true distances between cells [12].

In spite of these problems, the current practice in the field is to simply apply *ad hoc* clustering approaches to scRNA-seq data, typically inspired by unsupervised machine learning methods, with the aim of grouping cells of the same 'type', e.g. [13–15]. These clustering approaches generally include several complex layers of data pre-processing, such as normalization and imputation, feature selection, and dimensionality reduction, before the clustering algorithm is applied. These pre-processing steps not only include many fairly arbitrary choices but, as we have recently shown [12], such pre-processing can also severely distort the data by erroneously filtering true biological variability and introducing artefactual correlations. Furthermore, for the clustering itself many different approaches are available, and these typically additionally have many tunable parameters whose values in practice seem to be mostly set by trial-and-error. Given the many layers of *ad hoc* choices involved in these approaches, the resulting clusters lack any biophysical or even methodological interpretation. Instead, the approach taken to confirm the 'biological validity' of the clusters, is to show that the cluster exhibit some features that match known biological information, e.g. that certain 'marker' genes of a particular cell type are on average higher expressed in a given cluster. However, given that there are combinatorially many different clusterings that exhibit such partial matches with prior biological knowledge, it seems problematic to us to take such partial matches to prior biological knowledge as a validation of the clusters that happened to result from the complex layers of analysis that were applied to the data.

We strongly feel that, instead of applying *ad hoc* clustering methods and attempt to validate these retrospectively by comparison with prior biological knowledge, it is more constructive to first rigorously specify the aims of the analysis, and then *derive* the appropriate algorithm that accomplishes these aims from first principles. This is the approach we take here. We are not going to attempt to solve the general problem of how to define cell types and how to identify them, for reasons laid out above. Instead, our aim is to use clustering so as to maximally reduce the complexity of the dataset without losing *any* of the structure in the data. In particular, we aim to partition the cells of an scRNA-seq dataset into subsets such that the gene expression states of all cells within each subset are statistically indistinguishable. We thus aim to cluster cells at the highest possible level of resolution that is statistically meaningful, i.e. within each cluster all cells are within measurement noise in expression state, and between clusters the expression states are all distinct.

Because the nature of biological and measurement noise in scRNA-seq experiments is known, as characterized in previous studies [16], this task has a uniquely defined solution determined by first principles, as we show below. The resulting method, CELLSTATES, directly clusters the unnormalized data so that any pre-processing steps are avoided, measurement noise is properly taken into account, and there are no free parameters to tune. For example, the number of clusters is determined by the data, in contrast to most approaches in which the number of clusters is tuned by the user. Moreover, the resulting clusters have a clear and simple interpretation.

Because CELLSTATES only groups cells whose expression states are statistically indistinguishable, it typically divides the data into many more subsets than other clustering algorithms. To allow comparison with the more coarse clusterings provided by other methods, we additionally provide methods for hierarchically merging CELLSTATES's clusters into coarser clusters, to identify differently expressed genes below each branch point in the hierarchy, and to identify 'marker' genes that best distinguish between different clusters. As we show below, conventionally annotated biological cell types typically correspond to coarser clusters in this hierarchy, allowing us to interpret CELLSTATES's clusters as subtypes of conventionally annotated cell types.

## Materials and methods

### Multinomial noise in scRNA-seq data implies a parameter-free solution for probabilities of partitions of cells into states

The internal gene expression state (GES) of a cell $c$, which we will also refer to as a cellstate, is determined by a multitude of biological processes that influence the transcription rates $\lambda_{gc}(t)$ and degradation rates $\mu_{gc}(t)$ of mRNAs across genes $g$ and time $t$ in the history of the cell.

These rates determine the probabilities for the mRNA counts in the cell, which in turn ultimately determine the probabilities of the number of reads captured in a UMI-based scRNA-seq measurement. The probability distribution for the number of mRNAs in a cell $m_{gc}$ follows a Poisson distribution with mean $a_{gc}$ given by

$$a_{gc} \equiv \langle m_{gc} \rangle = \int_0^\infty dt \lambda_{gc}(t) \exp\left[-\int_0^t \mu_{gc}(s)ds\right], \qquad (1)$$

where the time is measured backwards from the present ($t = 0$) to the distant past ($t = \infty$) in the history of the cell [12]. Thus, notably, for each gene $g$ in each cell $c$, the entire complex history of transcription rate and mRNA decay rate can be summarized into a single parameter $a_{gc}$ that fully determines the probability distribution of its current mRNA count for gene $g$. The scRNA-seq measurement process is noisy and typically only a small fraction ($\sim$ 20% or less) of cellular mRNAs are captured. As this capture rate can vary substantially between cells, information about absolute gene expression levels is lost, at least to some extent. Therefore, more accurate inferences can be made regarding the expected *fractions* of total cellular mRNA that mRNAs of each gene $g$ represent. Following [12], we denote these fractions by transcription quotients $\alpha_{gc}$, which we define by

$$\alpha_{gc} = \frac{a_{gc}}{\sum_g a_{gc}}. \qquad (2)$$

We now define the GES of a cell as the vector $\vec{\alpha}_c$ of transcription quotients across all $G$ genes. Thus, a GES is a point in the $G$-dimensional simplex $\alpha_{gc} \geq 0 \ \forall g$ with $\Sigma_g \alpha_{gc} = 1$.

Given an scRNA-seq dataset with $N$ cells, we will assume that the GESs of the cells derive from an unknown set $S$ of GESs, where each GES $s \in S$ is characterized by a distinct vector of transcription quotients $\vec{\alpha}_s$. That is, we assume that there are somewhere between 1 (all cells having the same GES) and $N$ (all cells having a distinct GES) cellstates represented in the dataset. Our goal is to derive which cells are in the same state and thus separate differences in UMI counts due to biological and measurement noise from differences in the underlying biological state. Thus, the space of hypotheses for this problem is the space of possible partitions of the $N$ cells into non-empty non-overlapping subsets. In particular, we aim to calculate a likelihood for each possible partition that quantifies how probable the data is under the assumption that all cells in each subset of the partition are in the same GES.

The first step is to derive the relationship between the GES of a cell $c$ characterized by $\vec{\alpha}_c$ and the vector of its measured UMI counts $\vec{n}_c$, which is summarized in Fig 1A. Given the transcription activities $a_{gc}$ of a cell $c$, the mRNA counts are not uniquely determined, but due to inherent biochemical noise in the gene expression process, the mRNA counts $m_{gc}$ are given by Poisson samples with means $a_{gc}$. Defining the total transcription activity $A_c = \Sigma_g a_{gc}$, the expected mRNA count for gene $g$ can be expressed as the product of $A_c$ and the transcription

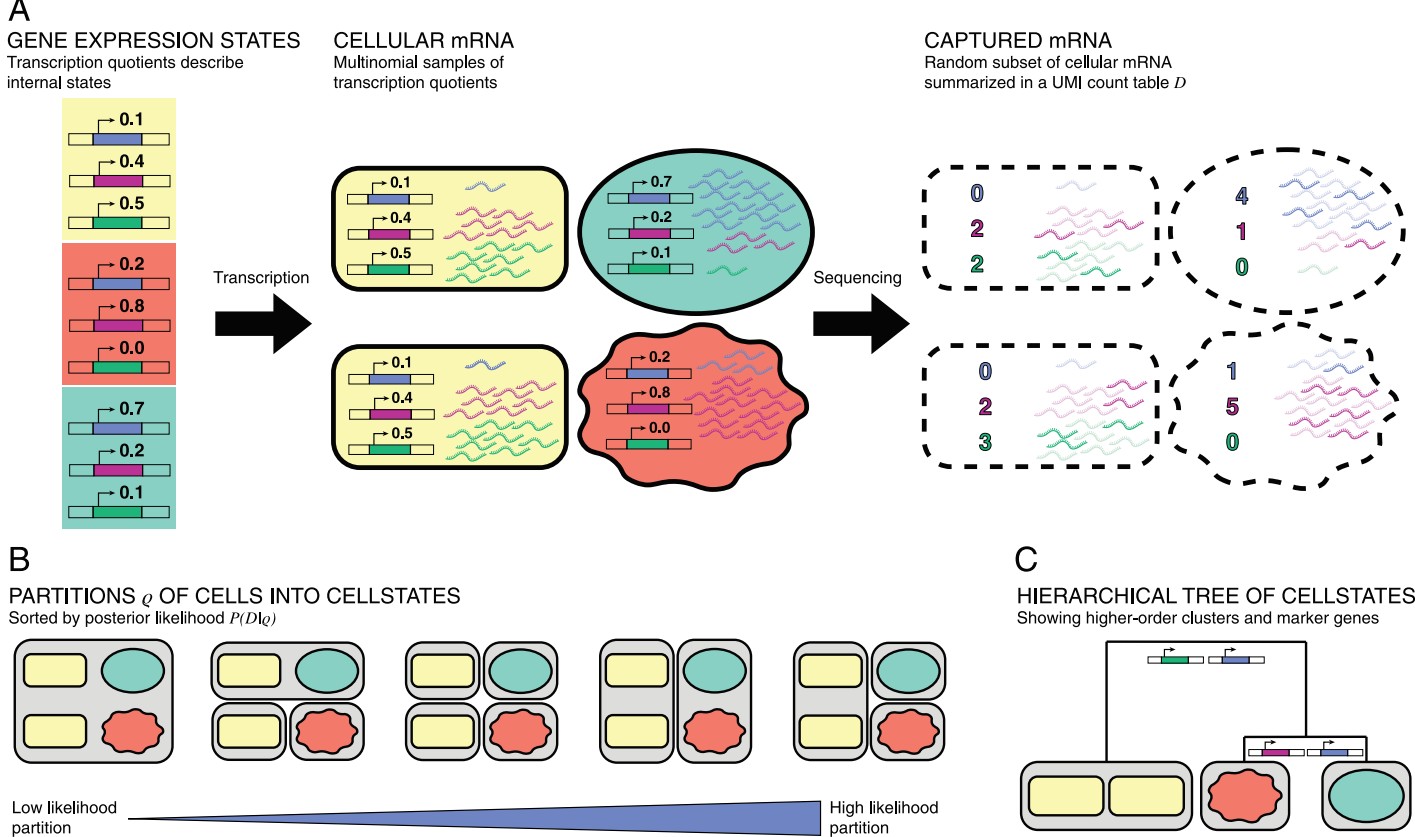

**Fig 1.** (A) Summary of model assumptions. Each cell is in a gene expression state (indicated by shape and color) characterized by the transcription quotients across genes. The relative numbers of mRNAs in the cell follow a multinomial distribution of these rates. The counts obtained from sequencing reflect a random subset of captured cellular mRNAs and follow the same multinomial. (B) Summary of clustering algorithm. Each partition $\rho$ of cells into clusters gives a likelihood of the data under the noise model. By optimizing the partition, we find groups of cells with shared gene expression states. (C) Cellstates can be hierarchically merged into higher-order cell types. For each merging step, we indicate which genes most contribute to distinguishing the transcription quotients to the left and right below the merger.

quotient $\alpha_{gc}$:

$$m_{gc}|\alpha_{gS}, A_c \sim \text{Poisson}(\alpha_{gc}A_c) \tag{3}$$

Assuming that, for cell $c$, each transcript was captured and sequenced with a probability $p_c$, the distribution of UMI counts $n_{gc}$ will also be Poisson distributed with mean $\alpha_{gc}A_cp_c$ for each gene $g$. If we marginalize over the unknown capture probability $p_c$ and condition on the total number of mRNAs $N_c$ that were captured for cell $c$, the counts $n_{gc}$ are simply distributed as a multinomial sample of the transcription quotients $\vec{\alpha}_{gc}$ (see the supplementary methods of [12] for a more extensive derivation):

$$\vec{n}_c|\vec{\alpha}_c, N_c \sim \text{Multinomial}(\vec{\alpha}_c, N_c) \propto \prod_g (\alpha_{gc})^{n_{gc}}. \tag{4}$$

Thus, the probability of the observed mRNA counts $\vec{n}_c$ of a cell $c$ conditioned on its GES $\vec{\alpha}_c$ is simply a multinomial sample of size $N_c$ of the expression state $\vec{\alpha}_c$. As an aside, we note that the vector of observed UMI counts $\vec{n}_c$ is the unique sufficient statistic for the GES $\vec{\alpha}_c$ of cell $c$.

Given a partition $\rho$ of the cells into non-overlapping subsets, we now use the above results to calculate a probability $P(n|\rho)$ of the observed UMI counts $n$ across all genes and cells, given the assumed partition $\rho$. The derivation of our model follows [17], is explained in detail in section A1 in S1 Text, and the general approach is illustrated in Fig 1. Briefly, a partition $\rho$ contains subsets of cells $s$, with one GES $\vec{\alpha}_s$ for each subset $s \in \rho$ (i.e. each subset $s$ corresponds to a cluster of cells), and all cells $c \in s$ are assumed to have the same GES $\vec{\alpha}_s$ for each subset $s \in \rho$. The probability for the counts $n_{gc}$ of all cells in the subset $s$ is simply the product over multinomial distributions for each of the cells. Thus, if we define the cluster UMI counts $n_{gs} = \sum_{c \in s} n_{gc}$ and $N_s = \sum_{c \in s} N_c$, then these cluster counts also simply derive from a multinomial distribution

$$\vec{n}_s | \vec{\alpha}_s, N_s \sim \text{Multinomial}(\vec{\alpha}_s, N_s) \propto \prod_g (\alpha_{gs})^{n_{gs}}. \tag{5}$$

Next, because we do not known the transcription quotients $\vec{\alpha}_s$ we marginalize over these parameters using a Dirichlet prior. The family of Dirichlet priors is the unique set of priors that is invariant under rescaling of the unknown transcription quotients $\alpha_{gc}$ and is parametrized by a vector of concentrations $\Theta$. This marginalization can be done analytically leading to a ratio of products of Gamma functions of the counts $n_{gs}$ (see section A1 in S1 Text). In this way, a likelihood of the UMI counts $\vec{n}_s$ is obtained for each subset $s$ in the partition $\rho$. By taking the product of these likelihoods over all subsets in $\rho$, we arrive at an expression for the likelihood $P(D|\rho, \Theta)$ of the entire dataset $D = \{\vec{n}_c\}$ as a function of the partition $\rho$ and the parameters of the Dirichlet prior $\Theta$. Taking a uniform prior over both partitions $\rho$ and the parameter $\Theta$, the posterior $P(\rho, \Theta|D)$ is simply proportional to the likelihood $P(D|\rho, \Theta)$, which we have obtained in analytical form. The aim of our algorithm is to now find the partition $\rho$ and prior parameters $\Theta$ that jointly maximize this likelihood.

Importantly, this approach uses only the assumptions that both the inherent biochemical noise in gene expression and the scRNA sequencing introduce Poisson sampling noise, and from first principles derives a parameter-free solution for the most likely partition $\rho^*$ of cells into cellstates that is entirely determined by the raw data $D$. Note that defining cellstates in this way also determines how many distinct cellstates there are, and how many cells there are in each state, directly from the data. The total likelihood of a partition $\rho$ simply quantifies how consistent the cells' measured UMI counts are with the assumption that all cells in each cluster share a common (but unknown) GES $\vec{\alpha}$. Importantly, over-clustering of the cells into too many cellstates is avoided through the Bayesian framework where increasing the number of GES $\vec{\alpha}$ that are marginalized over will lower the likelihood if not supported by the data. To summarize, we partition all cells into subsets such that it is most likely that within each subset the remaining variation between the captured UMI counts is due to random fluctuations.

## The likelihood function is optimized using a Markov-Chain Monte-Carlo algorithm

The number of possible partitions of $N$ cells grows faster than $e^N$, and we have confirmed that simple greedy searches, such as iteratively fusing clusters of cells to maximally increase the likelihood of the partition, tend to get stuck in local optima of the likelihood function. This makes maximization of the likelihood function challenging. To search for the optimal partition we start from the partition in which each cell forms a cluster by itself and use a stochastic Markov-Chain Monte-Carlo (MCMC) scheme as previously developed in [17]. In each step, a randomly selected cell is proposed to move into a randomly selected different cluster – and accepted if this move increases the likelihood of the partition. If the move decreases the likelihood by a factor $p < 1$, the move is accepted with a probability $\tilde{p}$ that is adjusted to ensure

uniform sampling of partitions (see section B in S1 Text). Although theoretically, this MCMC scheme samples partitions in proportion to their likelihood in the long run, we have observed that, in practice, either $p \gg 1$ or $p \ll 1$ for most of the proposed moves, most likely due to the fact that the total number of UMI per cell is generally large. Therefore, in practice the optimization essentially performs a random uphill walk to a local optimum rather than sampling the full probability distribution of partitions. We have experimented with a number of different search schemes, including simulated annealing and Gibbs' sampling schemes, but found that these random uphill walks provide the best balance between total run time and optimality of the final partition. After the MCMC converges, the optimization is followed by some deterministic steps as described in detail in section B in S1 Text. Multiple runs of CELLSTATES on the same data can yield slightly different partitions, and we simply select the best-scoring partition from the partitions obtained in different runs.

## Merging cellstates hierarchically into higher-order clusters

As the optimal cellstate partition gives a very fine-grained view of the data, it makes sense to relate the obtained cellstates to each other in a structured manner. To examine the higher-order structure between the cellstates of the optimal partition $\rho^*$, we devised a scheme to hierarchically merge them into higher-order clusters. We define a pairwise cluster similarity as the ratio of the likelihoods of the partition in which the two clusters are merged and the partition in which the two are separated. By construction of the optimal partition $\rho^*$, the similarity will be $< 1$ for any pair of clusters of this partition and we define a 'distance' between two clusters as minus the logarithm of this similarity. The most similar clusters are iteratively merged, resulting in a hierarchical tree of higher-order clusters, see section A3 in S1 Text and Fig 1C. As discussed below, we find that these higher-order clusters are often similar to the cell type annotations given in the publications of the datasets on which we ran our algorithm.

## Additional methods for cellstate annotation and running CELLSTATES incrementally

We have developed a number of additional tools to help users of CELLSTATES to explore and annotate their results, and to facilitate running of very large datasets by incrementally partitioning cells.

First, as described in section A4 in S1 Text, we developed a method that identifies genes that are most significantly differentially expressed between two sets of cellstates, e.g. the sets of cellstates on two sides of a branch point in the hierarchical tree of higher-order clusters. In particular, as explained in section A4 in S1 Text, the similarity of a pair of clusters (i.e. the ratio of likelihoods of the partitions with the clusters either merged or unmerged) can be approximated as the product of similarity scores for individual genes. This similarity score can thus be used to quantify the differential expression of a gene on opposite sides of a branch in the high-order cluster hierarchy and generally reflects the significance of the difference in *average* expression between the two sets of cellstates.

In addition, as described in section A5 in S1 Text, we developed another method for identifying genes that can best act as 'markers' to distinguish between two sets of cellstates. This method finds genes whose expression distributions have minimal overlap between the two sets of cellstates.

In order to annotate cellstates we also implemented tools that allows visualizing the composition of selected sets of cellstates in terms of the batches from which the cells derive, or any other annotation that is available for the cells.

Finally, CELLSTATES is computationally intensive and for very large datasets the runtimes can become unpractical (see Fig A in S1 Text for runtimes across the datasets analyzed in this work). First, for users that are restricted in consecutive runtime on their computational architecture, CELLSTATES allows for saving of intermediate 'breakpoints' and restarting the search of partitions from these breakpoints. Second, in order to make analysis of very large datasets possible, we have implemented a method that allows to run CELLSTATES incrementally. In particular, this method allows to add a set of cells to a previously obtained partition of another set of cells, and find an optimal joined partition. In this way, a large dataset can be divided into chunks of cells that are partitioned incrementally. Although this incremental running of CELLSTATES makes it less likely that the globally optimal partition is found, it allows to run very large datasets that otherwise would be impractical to run.

All these additional methods are provided on the Github page of CELLSTATES (see the Software Availability section below) where notebooks with example application of these methods are also provided.

## Results

### Cellstates accurately finds optimal partitions in simulated data

As discussed below, we tested CELLSTATES by running it on a number of published experimental scRNA-seq datasets. However, since there is no ground-truth information available for the GESs of cells in real scRNA-seq datasets, we decided to validate our likelihood maximization algorithm on synthetic datasets that were generated so as to be in agreement with the noise model described above. To get realistic simulated data, we modeled the simulations after results obtained by CELLSTATES from 18 of the analyzed real experimental datasets [2, 3, 18–27] as follows. For each of the 18 real datasets we took the optimal partition inferred by CELLSTATES and then for each of the clusters $s$ in this partition, sampled the UMI counts of its cells from a multinomial distribution with mean equal to the inferred GES $\vec{\alpha}_s$ of the cluster. We generated three independent simulated datasets for each of the 18 scRNA-seq datasets, for a total of 54 simulated datasets (See section B2 in S1 Text for details). We then ran the CELLSTATES algorithm three times on each simulated dataset. For the large majority of runs CELLSTATES found the exact same partition as the one that generated the data (Fig B in S1 Text), which is remarkable since only a tiny fraction of the total space of partitions is sampled during the MCMC likelihood optimization. Moreover, when the partition that CELLSTATES found differed from the partition that generated the data, this was because CELLSTATES found a partition with even higher likelihood than the one that generated the data, which occurs more frequently on down-sampled data (Fig C in S1 Text). In fact, for each of the 54 datasets our MCMC likelihood maximization procedure found a partition with likelihood at least as large as the partition that generated the data, and in $\approx 91\%$ of all runs overall (Fig B in S1 Text).

To compare the similarity of partitions more quantitatively, we will use the two complementary measures of homogeneity and completeness [28] throughout this paper. These measures quantify how much information (as quantified by the Gibbs/Shannon entropy function) a given partition $\rho$ contains about a reference partition $\rho_f$ and are both normalized to lie between 0 and 1. If we imagine that we color cells by their cluster in the reference partition $\rho_f$, i.e. so that all cells within each cluster of $\rho_f$ are given the same color, then homogeneity measures how much information the cluster membership in $\rho$ provides about the color of the cells (i.e. cluster membership in $\rho_f$). Homogeneity is 1 when, for each cluster of $\rho$, all cells have the same color. Completeness, vice versa, measures how much information the color of a cell provides about its cluster in $\rho$. Completeness is 1 when, for each color from $\rho_f$, all cells of a common color occur in only one cluster of $\rho$. Note that two measures are necessary because if $\rho$ is

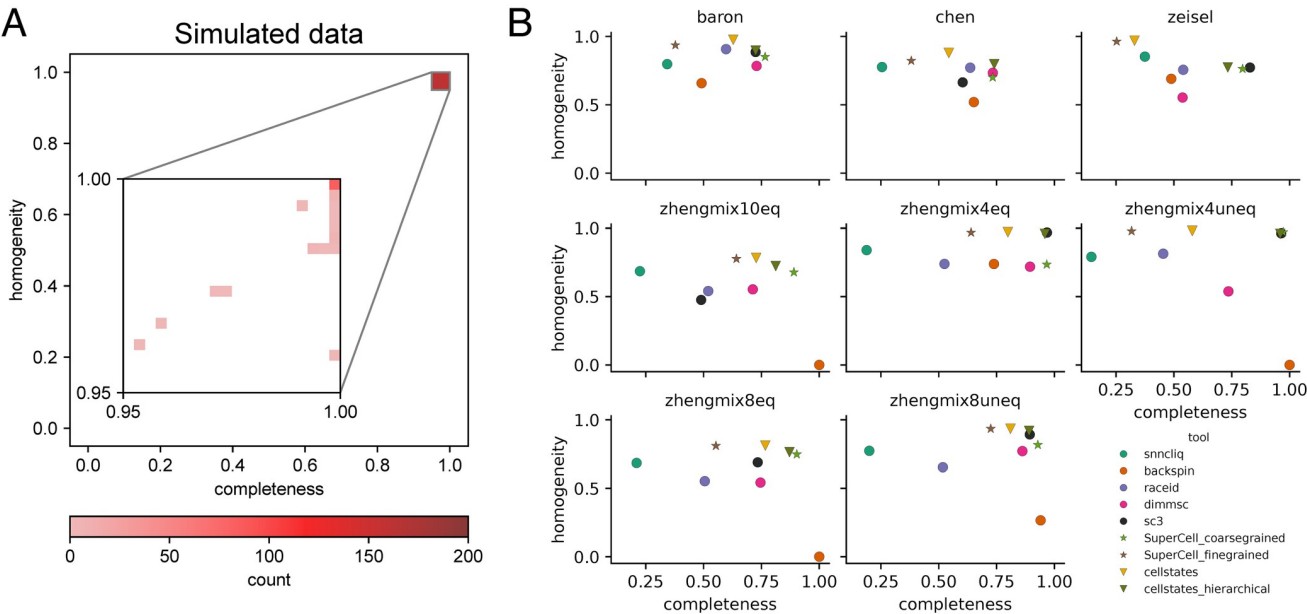

**Fig 2. Benchmarking of CELLSTATES.** (A) 2D-histogram of homogeneity and completeness of inferred cellstate memberships in 162 simulated datasets. The inset shows the distributions in the region [0.95, 1], [0.95, 1] where all results fall. (B) Comparison of the performance of CELLSTATES with those of other clustering tools. For CELLSTATES, we show the results for the full partition into cellstate-groups ("cellstates") and merged to the same number of clusters as the annotation ("cellstates_hierarchical"). For SuperCell we similarly show results with fine-grained setting matching the cluster number of CELLSTATES and with coarse-grained setting matching the cluster number of the annotation. In each plot, we show homogeneity and completeness of the partitions obtained by the different methods using the published annotation as the reference partitions. Note that low homogeneity and completeness may indicate partitions with, respectively, too few and too many clusters compared to the reference partition.

the partition in which each cell is its own cluster, homogeneity is 1 by definition (but completeness is 0). Vice versa, if $\rho$ is the partition in which all cells are in one cluster, than completeness is 1 (but homogeneity is 0).

Comparing the partitions inferred by CELLSTATES on the simulated data to the corresponding reference partitions used to generate the data, we find that they overlap very well, with completeness and homogeneity larger than 0.95 for all runs, and larger than 0.9975 for 118/ 162 (73%) of the runs, as shown in Fig 2A. As discussed in section B2 of S1 Text, most 'errors' occur when the maximum likelihood partition found by CELLSTATES is higher than that used to generate the data. This happens in particular when the simulated datasets are too noisy to resolve all ground-truth states because the total UMI counts of cells in the ground-truth states are smaller than in the original data.

In summary, our tests with simulated datasets show that on datasets that mimic real data, CELLSTATES performs extremely well on recovering the ground truth used to generate the data, most often recovering the exact partition. And when there is a difference in the partition found, this is most often because CELLSTATES found an even better partition, which is always very close to the ground-truth partition, as measured by completeness and homogeneity.

## Cellstates yields highly reproducible partitions on real datasets

We gathered a total of 29 published datasets from UMI-based scRNA-seq experiments, covering a large range of experimental protocols, tissues and two species (mouse and human), as summarized in S1 Table. We ran CELLSTATES five times on all datasets and compared the best-scoring partition from the five runs with the partitions from the other four runs. We find that

the agreement between multiple runs of CELLSTATES is high, with 88% of the homogeneity and completeness scores larger than 0.9 (Fig D of S1 Text). These results show that, even though different runs yield different partitions, the partitions found do not change substantially between runs.

## Cellstates partitions agree better with published annotations than those of other clustering tools

We next compared CELLSTATES partitions on real datasets with those of a set of previously clustering published methods (BACKSPIN [26], DIMM-SC [29], RaceID [30] and SC3 [31]), as well as a recent method that also aims to maintain most structure in the data by only clustering cells with highly similar gene expression states (SuperCell [32]). A short summary of these methods is provided in S2 Table. Assessing the relative performance of different clustering algorithms on real datasets is challenging because in general the ground truth is not known. Here we consider two tests. First, we selected 3 scRNA-seq datasets for which hand-curated annotations of cell types were provided in the publication [26, 33, 34] and compared the partitions obtained by each of the clustering methods with the published annotation. Although there is of course no guarantee that the published annotations are correct, it is reasonable to assume that a better match with these published annotations generally indicates better performance. Second, we also generated a set of five *in silico* mixtures of pure cell populations with known identity, as has been done before for similar benchmarks [16, 35], using data from [36], and checked to what extent each algorithm can correctly recover these known mixtures.

We ran each of the methods on each of these 8 datasets, without tweaking any of the default parameters (see section C in S1 Text for details). Only when a method required the cluster number to be set, we set it to the correct number of annotated cell types. For CELLSTATES we obtained both the partition given by the method itself, and the partition obtained when hierarchically merging clusters until the number of clusters matches the number of clusters in the annotation. To compare CELLSTATES directly with SuperCell, we ran SuperCell once with its graining parameter set 'finely' so as to produce the same number of clusters as CELLSTATES produced, and once more coarse grained so as to match the number of annotated clusters. As for our tests with the simulated data above, we compared the clusters obtained by each of the methods to the annotated clusters through homogeneity and completeness (Fig 2B).

Notably, the partitions found by CELLSTATES always had the highest homogeneity. This supports that CELLSTATES indeed only clusters together cells that are in the same underlying gene regulatory state and that this works automatically without the need to correctly set model parameters. Furthermore, although SuperCell with fine-grained settings produced partitions that have almost as high homogeneity on most datasets, its partitions have consistently lower completeness than those of CELLSTATES at the same cluster number.

CELLSTATES typically partitions the data in more clusters than the annotation, so that the completeness is generally significantly below one. When cellstates are hierarchically merged until the number of clusters matches the annotation (cellstates_hierarchical in Fig 2B), the completeness increases substantially, often without lowering homogeneity by much.

On the three datasets with published annotations, CELLSTATES obtained partitions that match the published annotations well, especially in comparison to the partitions produced by the other methods, with only SC3 and coarse-grained SuperCell showing a similar performance on these annotated datasets. Importantly, for the *in silico* cell mixtures—which are the closest we have to a ground-truth annotation—CELLSTATES clearly outperforms each other method on at least some datasets, and by a substantial margin across all datasets for all

methods except for SC3 and SuperCell (which match the performance of CELLSTATES on a sub-set of these datasets).

Overall, these results suggest that CELLSTATES can correctly predict higher-order cell types in scRNA-seq data and that it can do so more accurately than other tools on most datasets. Moreover, this performance is obtained without any need to pre-process the data or any adjustable model parameters.

## Cellstate diversity patterns depend on tissue of origin and not on technical features of the experiment

We next investigated the variation in the number and sizes of the clusters that CELLSTATES infers on different real datasets. Because measures such as the absolute number of cells per cluster obviously scale with the total number of cells sequenced, we decided to focus on the distribution of cellstate abundances $f_{\text{cellstate}}$, i.e. the fraction of cells associated with each cellstate. The distribution of $f_{\text{cellstate}}$ reflects the diversity of different GESs present in a given dataset. As an example, Fig 3A shows the distribution of $f_{\text{cellstate}}$ for data from the mouse cortex and hippo-campus of [26]. As illustrated by Fig 3A, we find that $f_{\text{cellstate}}$ typically varies over several orders of magnitude, and that a substantial fraction of the clusters correspond to singlets, i.e. where GESs were only associated with a single cell. That is, the counts in these cells are statistically different from those of all other cells. To obtain a quantitative measure of diversity we looked at various statistics of the distribution of $f_{\text{cellstate}}$ including the fraction of cells that are singlets, the average cellstate abundance $\langle f_{\text{cellstate}} \rangle$, its median, and the entropy of the distribution of $f_{\text{cellstate}}$.

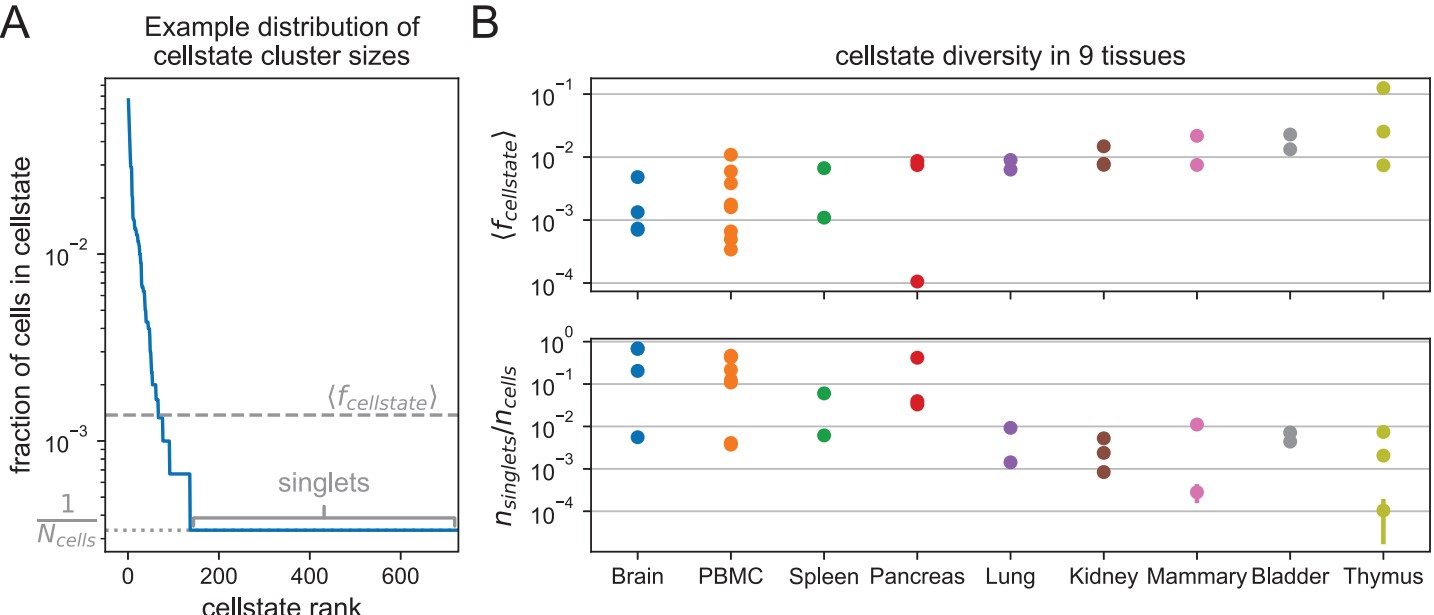

**Fig 3. Cellstate diversity reflects the tissue of origin of the data.** (A) Example rank-abundance curve for the fraction of cells associated with each cellstate in the dataset from [26]. Such curves describe the diversity of gene expression states in a dataset. The length of the horizontal tail gives the number of singlet cellstates $n_{\text{singlets}}$ with only one cell; the average cellstate abundance $\langle f_{\text{cellstates}} \rangle$ is also annotated. (B) For each of the CELLSTATES results for 29 different datasets from 9 tissues the average abundance $\langle f_{\text{cellstates}} \rangle$ and the fraction of singlets $n_{\text{singlets}}/n_{\text{cells}}$ are plotted by the tissue they originate from. These diversity measures show a clear dependence on the tissue, despite the large variation in experimental set-ups used. Error bars show the standard deviations from 5 independent runs of CELLSTATES and are often so small that they are not visible.

Although ideally these diversity measures would directly reflect the underlying biology of the tissue from which the data derives, we expected that these diversity measures might also strongly depend on technical features of the experiment such as the total number of cells and the typical total number of UMI per cell. For example, the higher the total UMI count per cell, the easier it becomes to distinguish subtly different cellstates, so that one would expect the cell-state diversity to increase with total UMI counts. Similarly, one would expect that the more cells are sequenced, singlet clusters should become less common. To investigate systematically to what extent the distributions of $f_{\text{cellstate}}$ reflect underlying biology versus technical features, we collected 29 scRNA-seq datasets from 9 different tissues, from different labs and using different sequencing technologies (see S1 Table), and investigated how the various diversity measures varied across tissues and with technical features such as cell number and total UMI counts.

Remarkably, we find that all these diversity statistics vary over several orders of magnitude across datasets. For example, the fraction of cells that are singlets varies over three orders of magnitude among the analyzed datasets, from $5 \times 10^{-4}$ to nearly $7 \times 10^{-1}$ (see Fig 3B), and similarly for the other statistics (Fig 3B and Fig E in S1 Text). Moreover, although there is a lot of variation across datasets, Fig 3B and Fig E in S1 Text also show that all the diversity measures systematically depend on the tissue of origin of the sample, despite vastly different experimental protocols used to obtain and sequence them. For example, for datasets stemming from biologically diverse cell populations such as brain or peripheral blood mononuclear cells (PBMC), CELLSTATES correctly and automatically infers few cells per GES, whereas vastly lower diversity is inferred for datasets stemming from Thymus. Moreover, and somewhat to our surprise, the diversity measures show almost no correlation with technical features such as total UMI counts and cell number (Fig F in S1 Text). The only clear correlation observed is a negative correlation between the number of cells and the median of the cellstate fraction. This correlation is explained by the fact that the median cellstate fraction often corresponds to singlets, i.e. $f_{\text{median}} = 1/n_{cells}$.

In summary, we find that the diversity of cellstates that is found in different datasets reflects the underlying biology of the system, and does not systematically depend on technical features of the experiment. Given this, the observation that in most of the datasets analyzed a large fraction of the clusters are singlets, strongly suggests that the true biological diversity of GESs is still severely under-sampled in these datasets. That is, many more cellstates exist than are captured at these sampling depths.

## Cellstates captures diversity of gene expression states in the mouse brain

Finally, to illustrate how the CELLSTATES data analysis pipeline can be used for in-depth analysis of a given dataset, we focused on the dataset of [26] consisting of 3005 cells from the somatosensory cortex and from the CA1 region of the mouse hippocampus. CELLSTATES infers a remarkable diversity in this tissue, with a total of 763 different GESs. Almost a quarter of the cells (727/3005) is in a unique singlet state, but there are also GES with up to 201 cells (7% of all cells), as can be seen in Fig 3A.

Visualizing how this large number of cellstates relates to another is difficult because the GESs are objects in a very high-dimensional space. The approach that is currently by far most popular in the field is to use stochastic embedding methods that attempt to place cells that are close in gene expression space near each other in a 2-dimensional plane, in particular UMAP [37] and t-SNE [38]. However, it is well appreciated that proper application of these tools is challenging [39], and that beyond a very rough approximate conservation of nearest neighbor relationships, all other structures in these visualizations are virtually meaningless. In fact, we

share the opinion of some in the field that the current usage of t-SNE and UMAP visualizations may be doing more harm than good [40]. Nonetheless, since such visualizations have become the *de facto* standard in the field, and anticipating the request to see where cells from identical cellstates map in such a visualization, we decided provide a t-SNE visualization of the data with cellstates annotated (Fig 4A, left panel).

This visualization shows that cells that are predicted by CELLSTATES to have the same underlying GES (indicated by the marker color, with singlets in gray) tend to be placed more closely in the t-SNE visualization. CELLSTATES infers that any variation between cells in the same GES is due to noise, and we can thus produce a simplified visualization by collapsing cells in each cellstate into a single disc at the average of their positions in the t-SNE visualization (right panel in Fig 4A, with the area of the disc corresponding to the number of cells in the cellstate). This illustrates CELLSTATES' ability to reduce the complexity in the data, allowing for a tidier visualization which, for example, suggests that different common cellstates (large colored discs) have different numbers of singlets (gray dots) in their neighborhood.

Next, we hierarchically merged the cellstates to determine higher-order clusters in the dataset, and again visualized the results by coloring either the cells or cellstates in the t-SNE visualization (Fig 4B). We see that when cellstates are merged into 8 higher-order clusters these clusters largely match the apparent clusters in the t-SNE visualization.

However, as it is virtually impossible to assign unambiguous meaning to the apparent structures in t-SNE visualizations, we feel that a more useful visualization of the relations between the cellstates is obtained by displaying the hierarchical tree resulting from iteratively merging the statistically most similar clusters (Fig 4C). The tree indicates which cellstates and higher-order clusters are most similar in expression, although it should be remembered that 'distance' between clusters is here measured in terms of how statistically significant the differences in the expression patterns are, as opposed to in terms of the magnitude of the changes in gene expression. Notably, at 8 higher-order clusters we find good correspondence with the cell type annotation given in the original publication (Fig G in S1 Text), and this is also confirmed by expression of marker genes for these annotated cell types (Fig I in S1 Text).

There are however two main differences. Firstly, at this level of resolution in our cluster hierarchy, the annotated clusters of endothelial-mural cells and microglia are merged and they separate only at 15 higher-order clusters (Fig H in S1 Text). Secondly, one cluster had a mixed annotation at the chosen resolution. As shown in Fig J in S1 Text, this cluster contains cells that express genes which are considered markers for multiple different cell types. This indicates that either the assumption that these genes are markers for specific cell types is incorrect or, alternatively, a technical artefact in the data, e.g. these 'cells' might correspond to multiple cells getting the same cell-barcode and having their counts combined.

We also provide software to extract genes that are most significantly differentially expressed between the cellstates on opposite sides of each branch. As explained in section A4 in S1 Text, the similarity of the gene expression states on opposite sides of the branch (calculated as the likelihood ratio of the partitions with the clusters merged versus unmerged) can be approximated as a product of similarity scores over genes and we use this similarity score to quantify differential gene expression. That is, genes that are most differentially expressed contribute most to the corresponding split in the tree.

To illustrate the use of this method, we focus on the cluster of interneurons, which was particularly diverse with 98 out of 290 cells in singlet states. For each node in the subtree corresponding to the interneurons we identified the genes that are most significantly differentially expressed between the cellstates at opposite branches below it, and in Fig 4D plotted the expression of these genes in a heat map with rows corresponding to genes and columns to individual cellstates. As expected, all columns display unique gene expression patterns,

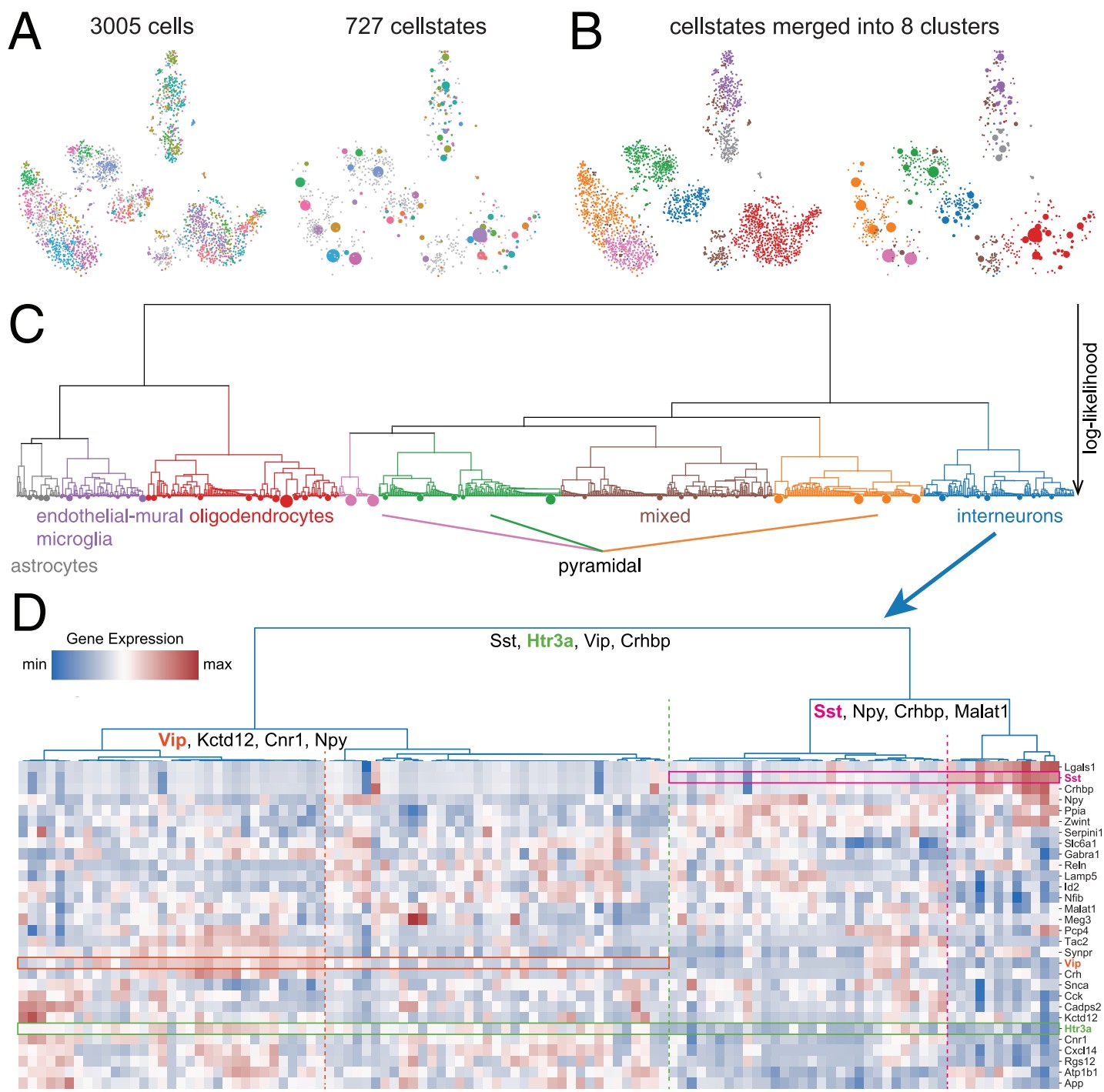

**Fig 4. Example analysis of a mouse cortex and hippocampus dataset [26] with CELLSTATES.** (A) Visualization of the data using t-SNE. The colors represent the inferred cellstates, with all singlets shown in gray. One the left, cells are shown individually while on the right cells in the same state were merged into discs. (B) The eight higher-order clusters shown in the same visualization as in (A), with colors representing the eight higher-order clusters defined in (C). (C) Hierarchical higher-order relations between the cellstates. Leaves of the tree correspond to cellstates with their area proportional to the number of cells in them. The vertical height of the branches indicates the negative log-likelihood of the corresponding partition. This tree allows us to split the data into eight higher-order clusters that correspond well to the cell types annotated in [26]. (D) Heat map of gene expression in the interneuron-cluster. Every column corresponds to one GES and shows the corresponding expression pattern. The hierarchical tree shown on the top corresponds to the interneuron sub-tree of (C). Rows correspond to selected genes that are differentially expressed between these GESs. In particular, for the first three splits in the tree, genes contributing most to their separation are indicated. Three of these are highlighted in the heat map in green, cyclamen, and orange.

confirming that there are clear differences between the GESs of all cellstates. Furthermore, for three nodes we highlight one example gene whose expression is clearly distinct between the corresponding branches by a rectangle in the heat-map, with the dotted line separating the cellstates on opposite sides of the branch.

At the highest level the gene *Htr3a* (green box in Fig 4D) is differentially expressed between interneuron cellstates to the left and right of the branch (dotted green line in Fig 4D). Similarly, the genes *Sst* (cyclamen box and dotted line) and *Vip* (orange box and dotted line) are differentially expressed between cellstates at branches lower in the tree of interneuron cellstate clusters. In general, we find many known markers of interneuron subtypes among the list of differentially expressed genes including *Sst*, *Npy*, *Crhbp*, *Cnr1*, *Cck* and *Vip* [26, 41–43] which supports the biological relevance of the cellstates that we identified.

It should be noted that the most significant genes identified using the method described in section A4 in S1 Text are those for which the *average* expression on opposite sides of the branch is most significantly different. However, since the expression of these gene might be quite variable across the cellstates below each branch, these differentially expressed genes are not necessarily optimal as 'marker' genes for distinguishing the cellstates on opposite sides of the branch. For this task, we developed a separate 'marker gene' method that identifies genes whose expression best distinguishes between two sets of cellstates (Suppl. Information section A5 in S1 Text).

The results on this dataset illustrate how CELLSTATES uncovers substantial sub-structure among cells of the interneuron type, with the tree structure illustrating the relationships between these subtypes, and the lists of top differentially expressed genes for each branch in the tree providing information regarding the biological differences between these subtypes.

Finally, one may wonder to what extent the sub-structures identified by Cellstates might reflect batch effects, i.e. whether different cellstates below higher-order clusters might simply correspond to cells from different batches. However, we find this not to be the case. In particular, the data of [26] derive from different samples of individuals of different age and sex and Fig K in S1 Text shows that all the larger interneuron GESs consist of mixtures of cells deriving from donors with different age and sex. In addition, we also used Cellstates to analyze a dataset of cells from the ventricular-subventricular zone (V-SVZ) of mouse brain that derive from 8 different samples of different sex and precise location (lateral and septal wall) [44]. Here too we find that individual cellstates tend to consist of relatively balanced mixtures of cells from different samples (Fig L and section D in S1 Text).

## Discussion

With the popularization of scRNA-seq a vast number of techniques for normalizing and post-processing single-cell gene expression profiles have been developed, including a large number of methods for clustering cells into 'cell types', e.g. as reviewed in [13, 35, 45, 46]. These methods typically involve several complex layers of analysis steps including the normalization of the raw data, transformation to logarithmic expression (or other 'variance stabilization' procedures), selection of 'features' to be used in the analysis, mapping of the data to a lower-dimensional representation, often involving abstract latent spaces, selection of a similarity or distance metric, and selection of the final clustering algorithm by which cells are grouped. Moreover, each of these analysis steps typically comes with tunable parameters.

Our impression of the current practice in the field is that the analysis methods are being deployed almost in a trial-and-error manner, i.e. with researchers iteratively trying out different methodologies and tweaking of their tunable parameters, comparing the results with expectations from prior biological knowledge, until results are obtained that look consistent

with prior knowledge and, ideally, make some new suggestions that appear biologically plausible.

We believe this kind of approach to analyzing complex data is extremely problematic. The layers of *ad hoc* processing steps and tweaking of parameters make it virtually impossible to give any unambiguous interpretation of the results, to rigorously compare results across different studies, and prohibits direct comparison of the results with those from other experimental approaches. Instead of iterative trial-and-error tweaking of several layers of *ad hoc* methods, we feel that the proper approach to data analysis is to specify the goals of the analysis and the assumptions about the data with enough precision, such that the proper analysis method is unique and transparently follows from these specifications.

This is the approach we have taken in this paper. Instead of attempting to solve the general and very difficult problem of how to determine which cells belong to the same 'type', we aimed to solve the simpler problem of maximally reducing the complexity of a given scRNA-seq dataset without *any* loss of structure, by grouping together all cells whose gene expression states are statistically indistinguishable. We have shown that, once a rigorous specification of the measurement noise relating the gene expression states of the cells to the raw scRNA-seq data is given, the appropriate clustering algorithm solving this problem is uniquely determined from first principles. We derived analytical expressions for the posterior probabilities for partitions of the cells into non-overlapping subsets in terms of the raw UMI counts across all genes and cells in the dataset, without any tunable parameters. Moreover, the clusters in the partition with maximal posterior probability have a clear and unambiguous interpretation: they are the optimal way of splitting cells into subsets with transcriptional profiles that are identical up to measurement noise.

A key assumption that our CELLSTATES algorithm makes is that the cells in a given dataset derive from a finite number of distinct gene expression states, i.e. that in general there will be groups of cells in identical gene expression states and one might wonder whether this is a realistic assumption. Indeed, given the complexity of the possible molecular internal states of cells, including the concentrations of all intracellular molecules, states of intracellular organelles, 3-D chromosome conformations, chromatin modifications, and so on, it seems unlikely that any two cells are in *exactly* the same state. Consequently, it seems *a priori* most reasonable to assume that the transcription quotient (TQ) vectors of cells derive from some unknown continuous density over the high-dimensional space of possible TQ vectors. However, since in an scRNA-seq experiment we only observe a finite sample of each cell's TQ vector, cells with sufficiently similar TQ vectors will be effectively indistinguishable. Thus, the assumption that our CELLSTATES algorithm makes boils down to assuming that the high-dimensional density over TQ space can be approximated by a set of local peaks where the TQ vectors in each peak are sufficiently similar so as to be statistically indistinguishable.

If instead of a collection of tight peaks, the density over TQ space would have extended continuous regions with similar density then one would expect the number of cellstates to grow systematically as the number of cells in the dataset increases. However, as we have seen in Fig F in S1 Text, we find that the observed cellstate diversity depends on the tissue of origin, and does not show systematic correlations with either number of cells or sequencing depth (i.e. total UMI count per cell). Although these are only fragmentary observations at this point and more in-depth study of this question is required, it hints that perhaps tight local peaks in TQ density do exist.

It should also be noted that most datasets analyzed here have a large fraction of singlet cellstates, i.e. clusters with only a single cell per cluster. This suggests that, for these datasets, we are still largely under-sampling the true diversity in cellstates that exist in most tissues and it is conceivable that for some tissues we might observe that the number of cellstates does continue

to grow as the total number of cells increases, which might then point to the existence of extended continuous regions with similar density of cellstates in TQ space.

One may also ask to what extent CELLSTATES' results are vulnerable to batch effects. Our measurement model makes some simplifying assumptions, such as ignoring potential systematic biases that cause transcripts of different genes to be captured with varying efficiency. We note that such gene-dependent capture efficiencies would not effect the distribution over partitions and the optimal partition $\rho^*$, as long as these capture biases are equal in all cells. In fact, as long as systematic biases are the same for the cells within each cellstate, the optimal partition would even remain the same if cells in different cellstates had different capture biases (see section A1 in S1 Text). However, all analysis of gene expression differences between different cellstates of course do rely on the assumption that capture biases are the same across all cells in a given dataset.

One limitation of our approach is that, since the number of partitions increases faster than exponentially with the number of cells and is vast for any realistic scRNA-seq dataset, there is no way to guarantee that our algorithm finds the global optimum even after re-running the program several times. However, our analyses of synthetic datasets with realistic size and structure shows that, in many cases, CELLSTATES manages to recover the single exact partition that generated the data, and when the generating partition was not recovered exactly, most often this was because a slightly different partition with even higher likelihood was found (Fig B in S1 Text). On real data we also found that the partitions obtained in different runs of cellstates are generally very similar (Fig D in S1 Text). These results suggest that the vast space of partitions can be effectively searched by CELLSTATES's Markov Chain Monte Carlo procedure.

It should be noted that, especially compared to most methods currently used in the field, on larger datasets CELLSTATES can have long run-times and requires significant computational resources, i.e. Fig A of S1 Text shows the runtimes of Cellstates on the datasets analyzed in this study, showing that although on average runtimes scale with the number of cells, different datasets with the same number of cells may vary 10-fold in their runtime. In the future we intend to improve the speed of the method by using computationally less expensive methods to either first subdivide larger datasets into coarse subsets before running CELLSTATES or to preselect neighborhood relationships, i.e. which pairs of cells are candidates for mergers with each other. For the time being, we have implemented a method to allow CELLSTATES to be run incrementally, i.e. to add a set of cells to an existing partition and find the optimal partition from this starting point. This facilitates analysis of larger datasets.

We also note that although it may take quite some time to run CELLSTATES on a dataset, it generally is still considerably less than the time required to perform the experiment. Moreover, since CELLSTATES has no tunable parameters, the method has to only be applied once. In fact, we believe that the strong importance that is currently assigned in the field to having fast analysis methods derives largely from the fact that most researchers apply these methods in a trial-and-error manner, running many times with different parameters settings and filters until results are obtained that 'look best' by some preconceived notions of what the data should show. As we already discussed above, we think this 'fast analysis' methodology is scientifically unhealthy, and like the movement advancing 'slow food' over 'fast food', we propose that analysis of complex large-scale datasets in biology would strongly benefit from a 'slow analysis' movement that favors slow but rigorously motivated methods over iterative tuning of fast *ad hoc* methods.

Finally, we would like to comment on the way we imagine CELLSTATES can be applied in practice. The most obvious application of CELLSTATES, and the one we highlighted here, is to identify subtle substructure among known cell types, the relationships between these subtypes, and the genes that most distinguish these subtypes. However, we feel that an arguable even

more important use of cellstates is as a way to significantly reduce the complexity of a dataset without losing *any* structure in the data. That is, after cells have been clustered into cellstates, one can decide to simply treat these clusters as if they consisted of a single cell that was much more deeply sampled and perform further analysis and processing such as trajectory reconstruction, pseudo-time analysis, visualizations or differential gene expression inference treating these clusters as if they were single cells, as has been proposed previously based on *ad hoc* clustering methods [47, 48]. CELLSTATES provides precisely the rigorous methodology for reducing the complexity of the dataset and removing some of the inherent noise in scRNA-seq data, while leaving all underlying biological variation completely intact. We propose that this application of CELLSTATES is an ideal first step in any scRNA-seq data analysis pipeline.

## Supporting information

**S1 Text. Supporting information file containing supplementary derivations, methods, text and figures.**
(PDF)

**S1 Table. Excel table with information on the datasets used.**
(XLSX)

**S2 Table. Excel table with information on other clustering tools.**
(XLSX)

## Acknowledgments

We thank van Nimwegen lab members for comments on the manuscript, Daan de Groot for identifying bugs in the code, and Thomas Julou for help with running SuperCell.

## Author Contributions

**Conceptualization:** Pascal Grobecker, Thomas Sakoparnig, Erik van Nimwegen.

**Data curation:** Pascal Grobecker, Thomas Sakoparnig.

**Formal analysis:** Pascal Grobecker, Thomas Sakoparnig, Erik van Nimwegen.

**Funding acquisition:** Erik van Nimwegen.

**Investigation:** Pascal Grobecker, Thomas Sakoparnig, Erik van Nimwegen.

**Methodology:** Pascal Grobecker, Thomas Sakoparnig, Erik van Nimwegen.

**Project administration:** Erik van Nimwegen.

**Software:** Pascal Grobecker, Thomas Sakoparnig, Erik van Nimwegen.

**Supervision:** Erik van Nimwegen.

**Validation:** Pascal Grobecker, Thomas Sakoparnig, Erik van Nimwegen.

**Visualization:** Pascal Grobecker, Thomas Sakoparnig, Erik van Nimwegen.

**Writing – original draft:** Pascal Grobecker, Erik van Nimwegen.

**Writing – review & editing:** Pascal Grobecker, Thomas Sakoparnig, Erik van Nimwegen.

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
