## [Decision Letter · Decision Letter 0]

16 Feb 2024

Dear Prof. van Nimwegen,

Thank you very much for submitting your manuscript "Identifying cell states in single-cell RNA-seq data at statistically maximal resolution" for consideration at PLOS Computational Biology.

As with all papers reviewed by the journal, your manuscript was reviewed by members of the editorial board and by several independent reviewers. In light of the reviews (below this email), we would like to invite the resubmission of a significantly-revised version that takes into account the reviewers' comments.

We cannot make any decision about publication until we have seen the revised manuscript and your response to the reviewers' comments. Your revised manuscript is also likely to be sent to reviewers for further evaluation.

Sincerely,

Ville Mustonen

Academic Editor

PLOS Computational Biology

William Noble

Section Editor

PLOS Computational Biology

Reviewer's Responses to Questions

**Comments to the Authors:**

Reviewer #1: The authors provide a method they term Cellstates to form clusters composed of cells that are statistically indistinguishable from each other, given noise from transcriptional bursting and random sampling of mRNA molecules during the experimental procedure of scRNA-sequencing. The method uses unnormalised scRNA-seq count data as input and is parameter-free. The method is computationally rather heavy and may not be suitable for larger data sets but as the authors openly state this and suggest improvements, it is not an issue for publication of the manuscript.

However, I have some issues with the basic concept and naming of the method which should be carefully addressed, as disclosed below (especially in points 1 and 2).

1. Even though methodologically different, the aim of the method is highly analogous to the MetaCell approach (Baran et al. Genome Res 2019; https://doi.org/10.1186/s13059-019-1812-2) that states in its abstract: “We describe a methodology for partitioning scRNA-seq datasets into metacells: disjoint and homogenous groups of profiles that could have been resampled from the same cell.“. Due to this overlap in the methods’ objectives, authors should add MetaCell to the set of benchmark methods (Fig 2). What is the advantage of Cellstates over MetaCell?

2. One major drawback for biologically meaningful usage of the method is the lack of addressing batch effects. The authors should show usage of the method on data compiled of several distinct batches (from distinct studies) of the same tissue to test whether there are any “cellstates” shared across the batches. If not, the term “cellstate” is misleading and should be replaced; for example the MetaCell concept discussed above is more realistic in this sense. ‘Cell state’ is a useful concept only if there are shared states across samples and tissues from distinct individuals; otherwise it would not be possible to make any generalizations of the functional potential of each state.

3. In the Discussion section, the authors suggest that as the “cellstate” number does not increase proportionally as the number of cells increases, cell states are in fact discrete rather than continuous. They also state that especially in more complex tissues, a large number of “cellstates” are formed by individual cells, suggesting that cell states are not sampled until saturation, as they also find larger cell clusters of statistically identical cells. However, the observation that the number of “cellstates” depends more on tissue (complexity) than on number of cells does not suggest against the continuum model. It could also be that the continuum is shorter for the less complex tissues, and that the continuums have more and less densely populated regions (some of which could be even extinct by the time of sampling). In this paradigm, the dense regions (analogous to valleys in Waddington landscapes) composed of larger number of statistically identical cells would be identified as “cellstates” as they are more likely to be sampled than sparser regions. Please improve the Discussion section on this matter.

4. “It should be noted that the most significant genes are those for which the average expression on opposite sides of the branch is most significantly different, but the expression of these genes might be quite variable across the cellstates below each branch.”

=> This pseudobulk type of approach of comparing averages can lead to rather spurious results, especially when considering that each “cellstate” can be composed of only a small number of cells. Please either justify this choice or improve the approach to extract markers?

Minor:

Please add citations to the referred 18 experimental datasets used as basis for the simulation to the Results where these data are first mentioned.

Reviewer #2: General Summary

Grobecker and van Nimwegen address the challenge of accurately identifying distinct gene expression states in single-cell RNA sequencing data. The authors discuss the limitations of existing methods, which often lack rigor and have multiple tunable parameters, making it difficult to assign concrete biophysical meaning to their results.

To overcome these challenges, the authors introduce a new tool, Cellstates, which partitions cells into subsets based on their gene expression states. This tool is designed to reduce the complexity of the dataset without eliminating meaningful structure, using a mathematically well-defined approach and operating without any tunable parameters.

The paper demonstrates that Cellstates can effectively identify subtle substructures within cell groups and that the diversity of gene expression states it identifies depends on tissue origin rather than technical experiment features. The tool also enables the placement of identified cell states into a hierarchical tree of higher-order clusters.

Overall, this paper proposes a novel and rigorous approach to analyzing single-cell RNA-seq data. The Python package is well documented in their GitHub repository, and it was straightforward to install their software on Linux, to run the tool using their command-line interface and to explore the results using the iPython notebooks they provide. The authors should be commended for this important resource to the community aiming to understand and interpret cellular diversity and gene expression states in different physiological, healthy and diseased contexts.

Technical Summary

In this paper, the authors introduce Cellstates, an extension of Sanity, introduced in Breda et al. 2021 (PMID: 33927416). In Sanity, it is assumed that each cell has a latent cell state, characterized by a simplex over all genes, called the transcription quotient (similar to how scVI models unique gene expressions). The observed UMI counts in each cell, given the unobserved (latent) transcription quotient, is modeled as r.v. with a Multinomial law. A major assumption is that the mRNA counts given the transcription quotient is independent from the other genes.

The contribution of this paper is to assume that distinct latent cell states explain the variance observed in the per cell UMI counts in each dataset. The partitioning cells is then based on the assumption that cells in the same cluster have the same latent cell state. For the cells in the same cluster, they assume a Dirichlet prior on the latent cell states, and, using the conjugacy of Dirichlet-Multinomial, marginalize out the latent cell states.

The goal of the inference is to find the most likely partition of the cell under this model.

They use empirical Bayes principles to set the parameters of the prior. Since the posterior distribution of the partitions and the scale parameter given the data does not have a closed form, MCMC is used to draw samples from it (iterating between sampling \\rho for a fixed \\Theta, and then updating \\Theta given the current \\rho).

In sum, their generative model is as follows:

- Draw a partitioning \\rho

- For each partition S \\in \\rho:

* Draw \\alpha_S ~ Dirichlet(\\Theta \\phi)

- For each cell c in partition S:

* Draw total UMI count N_c

* Draw UMI counts n_c ~ Mult(\\alpha_S, N_c)

Comments

We have several comments for the authors to address:

- The authors should qualify their statements about batch effects in Section 3.4 and in the Discussion further. While they report that cellstate diversity reflects inter-tissue variability rather than technical variability in Section 3.4, they should expand their analysis to consider inter-donor variation within each of their datasets.

- The authors should validate Cellstates on a wider range of datasets. The paper primarily focuses on the application of the Cellstates tool on specific datasets. Expanding this analysis to a wider variety of datasets, especially those with varying complexities, could enhance the generalizability and robustness of the conclusions. Specifically, there are several common scenarios in scRNA-seq data analysis where Cellstates could identify subtle substructure within cell types, which would exemplify its relevance to a broader readership. One such setting is in cancer datasets, where malignant cells are often aneuploid and can result in correlated changes in expression of neighboring genes that are simultaneously affected by copy number aberrations. These typically result in donor-specific gene expression states, and thus Cellstates may be well positioned to directly identify such states. There is a vast literature reporting scRNA-seq datasets in genomically unstable cancers where they could find some test datasets to try this (e.g. PMID: 36517593, PMID: 29674595). Another such setting is in the study of immune dysfunction, which has been explored in the context of aging, infection and cancer. Specifically, the progression of T cells across a phenotypic continuum traversing discrete cell states (naive, effector/memory, pre-dysfunctional, dysfunctional) is hotly debated in the field. There is also substantial literature on this topic where they can find example datasets (e.g. PMID: 30595452).

- The authors advocate for slow and rigorous data analysis, which we fully agree with. Given the increase in our ability to profile hundreds of thousands to millions of cells by scRNA-seq nowadays however, they should include results of the runtime on their benchmarking datasets with variable number of cells. It would be great if they could also include some recommendations for users aiming to apply their tool to large datasets, who may for example resort to analyzing subsets of a dataset with Cellstates which may not guarantee the same partitions.

- Figures S8 and S9 are missing a color scale.

- It would be helpful to add a table in Supplementary A.1. where all random variables (r.v.s) and their respective symbols are defined.

- Also, it will help the readability of the methods section to have a table where the probability distribution function of these r.v.s. is noted.

- It appears that the parameters of the prior are learnt in a data-dependent manner (\\Theta, the scale factor, and \\psi, the simplex.) This is a sensible approach; however, it means that the method falls under the empirical Bayes umbrella. This should be explicitly noted and the relevant papers cited.

- The writing of the computational section A.2. should be improved. The target posterior should be clearly indicated (P(\\rho , \\Theta | D) ?). Then the fact that it is interactable, and therefore MCMC (Metropolis within Gibbs) is used to sample from this posterior.

**Have the authors made all data and (if applicable) computational code underlying the findings in their manuscript fully available?**

Reviewer #1: Yes

Reviewer #2: Yes

PLOS authors have the option to publish the peer review history of their article (what does this mean?). If published, this will include your full peer review and any attached files.

Reviewer #1: No

Reviewer #2: **Yes: **Ignacio Vazquez-Garcia

Figure Files:

Data Requirements:

Please note that, as a condition of publication, PLOS' data policy requires that you make available all data used to draw the conclusions outlined in your manuscript. Data must be deposited in an appropriate repository, included within the body of the manuscript, or uploaded as supporting information. This includes all numerical values that were used to generate graphs, histograms etc.. For an example in PLOS Biology see he

---

## [Decision Letter · Decision Letter 1]

4 Jun 2024

Dear Erik,

Please act on the request by Referee 2: *"However, we still ask that the authors add a table explaining their notation in Supplementary A.1."*  while completing other final formatting requests. 

Best wishes,

Ville

Dear Prof. van Nimwegen,

We are pleased to inform you that your manuscript 'Identifying cell states in single-cell RNA-seq data at statistically maximal resolution' has been provisionally accepted for publication in PLOS Computational Biology.

Best regards,

Ville Mustonen

Academic Editor

PLOS Computational Biology

Jian Ma

Section Editor

PLOS Computational Biology

Reviewer's Responses to Questions

**Comments to the Authors:**

Reviewer #1: The authors have addressed all my comments and from my perspective the manuscript is now ready for publication.

Reviewer #2: We thank the authors for their thoughtful replies to our comments, as well as to Reviewer 1’s comments. The authors extended their initial presentation of the paper, including expanded benchmarking of the Cellstates method against additional existing methods, analyzing cellstate variation in the context of batched experimental designs, and incorporating new software features into Cellstates to derive cellstate marker genes and facilitate analysis of large datasets.

We have a minor point to follow up on, which if fulfilled would be sufficient for paper acceptance. The comment below is numbered in the same order as the authors’ replies.

Comment 2.5: We appreciate the authors’ sense of humor. We do not intend to ask the authors to use our toothbrushes or to impose a particular choice of notation. However, we still ask that the authors add a table explaining their notation in Supplementary A.1.

**Have the authors made all data and (if applicable) computational code underlying the findings in their manuscript fully available?**

Reviewer #1: Yes

Reviewer #2: Yes

PLOS authors have the option to publish the peer review history of their article (what does this mean?). If published, this will include your full peer review and any attached files.

Reviewer #1: **Yes: **Anna Vähärautio

Reviewer #2: **Yes: **Ignacio Vazquez-Garcia

---

## [Editor Report · Acceptance letter]

5 Jul 2024

PCOMPBIOL-D-23-01798R1 

Identifying cell states in single-cell RNA-seq data at statistically maximal resolution

Dear Dr van Nimwegen,

I am pleased to inform you that your manuscript has been formally accepted for publication in PLOS Computational Biology. Your manuscript is now with our production department and you will be notified of the publication date in due course.

With kind regards,

Zsuzsanna Gémesi
